# Genetic Diversity and Population Structure of African Sorghum (*Sorghum bicolor* L. Moench) Accessions Assessed through Single Nucleotide Polymorphisms Markers

**DOI:** 10.3390/genes14071480

**Published:** 2023-07-20

**Authors:** Muhammad Ahmad Yahaya, Hussein Shimelis, Baloua Nebie, Chris Ochieng Ojiewo, Abhishek Rathore, Roma Das

**Affiliations:** 1African Centre for Crop Improvement, School of Agricultural, Earth and Environmental Sciences, College of Agriculture, Engineering and Sciences, University of KwaZulu-Natal, Private Bag X01, Scottsville, Pietermaritzburg 3209, South Africa; 2Department of Plant Science, Institute for Agricultural Research Samaru, Ahmadu Bello University Zaria, PMB 1044, Kaduna 810211, Nigeria; 3International Maize and Wheat Improvement Center (CIMMYT), P.O. Box 3320, Escale Thiès BP 3320, Senegal; baloua.nebie@cgiar.org; 4International Maize and Wheat Improvement Center (CIMMYT), ICRAF House, United Nations Avenue, Gigiri, P.O. Box 1041, Nairobi 00621, Kenya; c.o.ojiewo@cgiar.org (C.O.O.); abhishek.rathore@cgiar.org (A.R.); r.das@cgiar.org (R.D.)

**Keywords:** accessions, population structure, gene flow, single nucleotide polymorphism, *Sorghum bicolor*

## Abstract

Assessing the genetic diversity and population structure of cultivated sorghum is important for heterotic grouping, breeding population development, marker-assisted cultivar development, and release. The objectives of the present study were to assess the genetic diversity and deduce the population structure of 200 sorghum accessions using diversity arrays technology (DArT)-derived single nucleotide polymorphism (SNP) markers. The expected heterozygosity values ranged from 0.10 to 0.50 with an average of 0.32, while the average observed heterozygosity (0.15) was relatively low, which is a typical value for autogamous crops species like sorghum. Moderate polymorphic information content (PIC) values were identified with a mean of 0.26, which indicates the informativeness of the chosen SNP markers. The population structure and cluster analyses revealed four main clusters with a high level of genetic diversity among the accessions studied. The variation within populations (41.5%) was significantly higher than that among populations (30.8%) and between samples within the structure (27.7%). The study identified distantly related sorghum accessions such as SAMSORG 48, KAURA RED GLUME; Gadam, AS 152; CSRO1, ICNSL2014-062; and YALAI, KAFI MORI. The accessions exhibited wide genetic diversity that will be useful in developing new gene pools and novel genotypes for West Africa sorghum breeding programs.

## 1. Introduction

Sorghum [*Sorghum bicolor* (L.) Moench, 2n = 2x = 20] is the fifth most important cereal crop in the world after maize, wheat, rice, and barley. The global production area of sorghum is close to 50 million hectares, with production levels exceeding 60 million tons per annum [1]. Sorghum is relatively more tolerant to biotic and abiotic stresses than other cereals, resulting in its wide adaptation, leading to production in marginal conditions across the globe. In these agro-ecologies, sorghum production is more dependable than other cereal crops such as wheat, rice and maize [2]. These characteristics have contributed to sorghum being the staple food crop for more than 500 million resource-poor people in more than 98 countries in the semi-arid and arid regions of Africa and Asia [3]. Nigeria grows approximately 40% of the total sorghum production in Africa (6.9 million tons) and is the second largest global producer after the USA (9.24 million tons). However, the area planted with sorghum in Nigeria (5.82 million ha) is more than double the sorghum production area in the USA (2.04 million ha) [1]. However, productivity in the USA is at 4.5 t ha^−1^, while the productivity of sorghum in Nigeria is trailing at 1.2 t ha^−1^. A similar trend has existed for the past three decades in West Africa, where the area planted with sorghum has increased by 50% but yields an average of <1 t ha^−1^ [4]. The farmers in Nigeria and West Africa mainly cultivate unimproved landraces that are low yielding but are adapted to thrive under the region’s harsh environmental conditions, generating reliable harvests [4].

Several constraints, including drought, low soil fertility, *Striga*, and stem borer disease, limit sorghum productivity in Africa [4]. Fortunately, there is a broad diversity of landraces mainly grown by farmers and used for sorghum improvement in the region in response to these different constraints. The landraces are adapted to harsh conditions, making them vital genetic resources, possessing stress tolerance genes that can be exploited in sorghum breeding programs. The Institute for Agricultural Research (IAR) Samaru, Nigeria, maintains the most extensive *ex-situ* sorghum germplasm collection among the National Agricultural Research Systems (NARS) in West Africa. The germplasm collection is a priceless genetic resource for various breeding programs, nationally and internationally. To fully utilize these genetic resources in gene banks, breeders should explore the genetic variation within and among the collections and accessions. This is necessary for efficient use of genetic resources and resource allocation to breeding projects and to minimize handling of duplicated accessions or closely related accessions. At present, the genetic diversity and genetic structure of IAR’s conserved sorghum germplasm have not been fully documented. Thus, there is a need to assess the underlying genetic diversity and structure in the germplasm to devise optimal breeding strategies for sorghum at IAR.

Genetic diversity in germplasm collections is routinely assessed using different phenotypic and molecular markers. Molecular markers have been extensively used in genetic diversity studies because they are not affected by changes in environmental factors. Several molecular marker technologies have been developed, including proteins [5], random amplified polymorphic DNA (RAPD) [6], restriction fragment length polymorphism (RFLP) [7], inter-simple sequence repeat (ISSR) [8], amplified fragment length polymorphism (AFLP) [9], simple sequence repeat (SSR) [10], expressed sequence tag-Simple Sequence Repeat (EST-SSR) [11] and single nucleotide polymorphism (SNP) [12,13]. Each of these markers has its advantages and limitations, including low marker density, inadequate genome coverage, or cost per sample.

Diversity Array Technology (DArT) was developed in early 2000 to minimize a bottleneck inherent to other marker platforms. The DArT platform utilizes a microarray hybridization method to produce thousands of polymorphic loci in a single assay. The platform is fast becoming a marker of choice because it provides a cost-effective sequencing that is independent of prior sequence information with ultra-high-throughput marker systems. DArT markers have been used successfully in population genetic studies of sorghum [14]; but also, in other crops such as barley [15]; wheat [16]; macadamia [17] and maize [18]. Cuevas et al. [10] and Girma et al. [19] assessed genetic diversity in sorghum using SNPs and found high levels of differentiation among the Ethiopian accessions they studied.

Conversely, Lasky et al. [20] reported less genomic variation among sorghum accessions from East Africa when assessed using SNPs. The differences in genomic variations can be attributed to genotype differences, the autogamous nature of sorghum, the accumulation of local diversity over time and more recombination events that break linkages between adaptive and neighbouring loci. Information on population genetic structures and familial relatedness among sorghum accessions in West Africa is lacking based on reliable marker systems such as SNPs. Therefore, the objectives of this study were to assess the genetic diversity and deduce the population structure among 200 sorghum accessions. The accessions form part of a core collection of germplasm used as parental lines in several sorghum breeding programs in Nigeria and neighbouring countries. The information generated will be valuable for sorghum pre-breeding by identifying diverse parental germplasm for core breeding.

## 2. Materials and Methods

### 2.1. Germplasm

Seeds of 200 sorghum accessions from an existing collection (Appendix A) were obtained from national and international research Institutes; 130 landraces from Institute for Agricultural Research (IAR) Samaru, Nigeria, 60 elite breeding lines and landraces from the International Crops Research Institute for the Semi-Arid Tropics (ICRISAT) Kano Station Nigeria, and 10 lines from the African Centre for Crop Improvement (ACCI) South Africa used in this study. The selected accessions from IAR and ICRISAT were collected in farmers’ fields in 2018 from sorghum growing regions and agro-ecological zones in Nigeria and are landraces that are well adapted to the environmental conditions and agricultural practices under which they are grown. In addition, the ACCI collections are accessions adapted to the growing agro-ecologies in South Africa.

### 2.2. DNA Extraction and Genotyping-by-Sequencing (GBS)

The genotypes were grown in a plant growth chamber (Conviron, Canada) at the Biosciences eastern and central Africa-International Livestock Research Institute (BecA-ILRI) hub using cell trays. Three seeds of each genotype were sown per tray. Three-week-old leaf sample was collected from the three seedlings and the pooled leaf samples were frozen in liquid nitrogen and stored at −80 °C for later use. Genomic DNA (gDNA) was extracted from the frozen tissue according to the CTAB protocol, with some modifications [21]. The quantity of extracted DNA in each sample was determined using a Thermo Scientific NanoDrop Spectrophotometer 2000 c. The quality of the extracted DNA was checked on 0.8% agarose gel run in 1% TAE buffer at 70 V for 45 min. After the quality had been checked, 40 μL of a 50 ng/μL gDNA of each sample of the 200 sorghum lines was sent for whole genome scanning using Genotyping by sequencing (GBS) technology as described by Elshire et al. [22], using DArTseqTM technology (https://www.diversityarrays.com/ (accessed on 22 February 2022) of the Integrated Genotype Service and Support (IGSS) platform in Nairobi, Kenya. The GBS was performed by using a combination of DArT complexity reduction methods and next generation sequencing following protocols described in [23,24,25]. Marker development was based on the protocol of Elshire et al. [22], using the ApeKI restriction enzyme (recognition site, G| CWCG). Reads and tags found in each sequencing result were aligned to the sorghum reference genome v2.1 (available via https://phytozome.jgi.doe.gov/pz/portal.html#!info?alias=Org_SbicolorRio_er (accessed on 5 April 2022). Each allele was scored in a binary fashion (“1” = Presence, “0” = Absence and ‘-’ for failure to score) while heterozygotes were scored as 1/1 (presence for both alleles/both rows).

### 2.3. Data Analysis

Genetic diversity parameters: polymorphic information content (PIC), minor allele frequency (MAF), major allele frequency (MaF), observed heterozygosity (Ho), and expected heterozygosity or gene diversity (GD) were estimated for each defined group with R software [26].

### 2.4. Population Structure Analysis

Two complementary methods were employed to deduce the population structure of the 200 African sorghum accessions. The first method involved using a Bayesian model-based clustering algorithm called STRUCTURE software [26,27]. The second method employed principal coordinate analysis (PCoA). Bayesian-based clustering was performed using STRUCTURE software v.2.3.4 (Pritchard Lab, Stanford University, Stanford, CA, USA) [28] with four independent runs with K from 2 to 10, each run with a burn-in period of 10,000 iterations and 50,000 Monte Carlo Markov iterations, assuming the admixture model. The output was subsequently visualized using the web-based program Structure Harvester v.06.94 [28] and the number of clusters was inferred according to the Evanno method [29]. The software CLUMPP version 1.1.2 (Rosenberg lab, Stanford University) [30] was utilized to align the cluster assignments obtained from separate runs. The input files generated by structure Harvest were employed for this purpose. Bar plots were created using DISTRUCT version 1.1 (Rosenberg lab, Stanford University) [31] to visualize the average outcomes of the runs, focusing on the most likely value of K. A genotype was considered part of a particular group if its membership coefficient was ≥0.70. Genotypes with membership coefficients below 0.70 for each assigned K were considered to be admixed. Phylogenetic relationships between the lines were inferred using the unweighted neighbour-joining method [32] and plotted using R software [28] based on Rogers’ dissimilarity [33]. Heatmap of the genetic distance value among lines was generated using gplots R package. The adegenet R package [34,35] was used to extract and plot pairwise distances between different groups identified from structure analysis. PCoA (Principal Coordinates Analysis) is a method that uses distances to analyze and visualize differences between individuals. The Eigenstrat method [36] based on principal components analysis was used to study population relationships further, and two-dimension PCA plot was generated using ggplot2.

The number of subpopulations determined with STRUCTURE was used for analysis of molecular variance (AMOVA) and the calculation of Nei’s genetic distance [37] using poppr package in the R version 2.8.3 [28,37]. From AMOVA, the fixation index (FST) and Nm (haploid number of migrants) within the population were obtained. FST measures the amount of genetic variance that can be explained by population structure based on Wright’s F-statistics [37], while gene flow was estimated using an indirect method based on the number of migrants per generation (Nm) as (1 − FST)/4 FST.

The phylogenetic relationships of the subpopulations were generated based on pair-wise fixation indexes using the StAMPP package [38] and neighbor joining trees were constructed using the dartR package in R. The dissimilarity or distance matrix, based on pair-wise genetic frequencies, was computed using the Euclidean distance formula in the R environment. The resulting dissimilarity matrix was used to create a tree structure using the unweighted pair group method analysis (UPGMA) algorithm, which was performed using the ggdendro and ggplot2 packages within the same software. Phylogenetic trees were constructed in R using the hclust algorithm, and the UPGMA agglomeration method was employed to determine the relevant clustering of data points.

## 3. Results

### 3.1. Genetic Parameters

The summary statistics of the 7516 SNP markers are presented in Table 1. The collection exhibited important diversity, with a gene diversity (GD) values ranging from 0.10 to 0.50 with an average value of 0.32. The heterozygosity (He) value ranged from 0.01 to 0.79 with an average value of 0.15. In connection, the average inbreeding coefficient (F_IS_) of 0.53 was moderate. From the DArT SNP markers, Appendix A showed that 30.7% of the SNP markers had a PIC ≤ 0.20, 27.6% had a PIC ≤ 0.29, and approximately 41.6% had a PIC ≤ 0.37. Measures of Minor allele frequency (MAF) ranged from 0.05 to 0.50, with a mean of 0.23. The major allele frequency (MaF) was 0.50 for minimum and 0.95 for maximum, with an average of 0.77 (Table 1).

### 3.2. Population Structure and Genetic Relationships

The population structure of the 200 sorghum accessions revealed four distinct subpopulations (Figure 1A,B). The number of clusters (K) was plotted against ΔK which revealed the highest peak to occur at K = 4 (Figure 1A) and each genotype was assigned to a cluster (represented by different colours in Figure 1B). The list of genotypes and the overall representation of membership of the sample in each of the four clusters are presented in Appendix A. The optimal K value suggests that four groups (G1, G2, G3, and G4) revealed the highest probability for population clustering, which consisted of 50, 49, 48, and 53 accessions, respectively (Figure 1B). Group G1, comprising 25% (50 accessions) of the collection, included drought-tolerant accessions from ICRISAT and landrace collections from IAR obtained from local farmers in Nigeria. The G2 group comprised 24.5% of the population (49 accessions) of local landraces and elite breeding lines from ACCI, ICRISAT, and IAR. The diverse membership from different sources of collections containing local landraces and improved cultivars suggested a shared ancestry. In addition, Groups G3 and G4 consisted of 24% (48 accessions) and 26.5% (53 accessions) of the total population, respectively (Appendix A). They comprised of landraces from IAR and ICRISAT obtained from local farmers in Nigeria and improved sorghum cultivar (improvement of local landraces by introgression with introductions obtained from ICRISAT by IAR). The pairwise genetic distances of the four subpopulations identified in STRUCTURE revealed members from the same group were closer than those from different groups (Figure 2).

Based on the pairwise genetic distance matrix among all the 200 accessions, the principal coordinate analysis (PCoA) revealed four clustered groups in accordance with the STRUCTURE results (Figure 3). The total amount of genetic variation explained by the first two principal coordinates was 25.1%. The PCoA clearly separated Groups G1 and G4 by PC2, showing a higher admixture between ICRISAT and IAR collections. The other two Groups, G2 and G3 were distributed along PC1, including 2 members from G1. Although some degree of overlap among G1 and G4 gene pools appeared at the center of PC2 quadrant, there was no apparent overlap in PC1 with G2 members located at the upper extreme of PC1 while G3 members were distributed along the lower extreme of PC1. From the results of the PCoA, the groups G1 and G3 distributed along the lower and upper extremes of PC1 were the most distant of the four groups and comprised of all the collections in ACCI in G2 and the majority of the collections from ICRISAT and IAR in G3. Overall, there was a high level of overlap between the ACCI, IAR and ICRISAT sorghum accessions.

The genetic distance among the population is represented by the neighbor-joining phylogenetic tree (Figure 4A). The neighbor-joining phylogenetic tree grouped the 200 accessions into four significant clusters in concordance with the STRUCTURE (Figure 1B) and PCoA (Figure 3) results with high degrees of admixture among the sources of collection. When the neighbor-joining tree was performed for the accessions according to their biological race (Figure 4B), there was no clustering in accordance with race. However, collections from IAR (especially the improved and released cultivars such SAMSORG 43) were found to be interspersed with all the collections from ACCI. Finally, other materials from IAR and ICRISAT seem to have been obtained from the common landraces grown in Nigeria i.e., Kaura and Fara-fara varieties and they form different clusters from those introductions used in breeding programs. The Kaura variety are derived from mostly Durra–Caudatum races while the Fara-fara variety are derived mostly from Guinea–Caudatum races. As expected, they form the same clusters with their derivatives and different clusters from the other groups. From the results, the phylogenetic tree revealed the branching history of common ancestry of the accessions under study.

### 3.3. Analysis of Molecular Variance (AMOVA)

The four subpopulations identified in STRUCTURE were applied in R software to calculate the AMOVA, fixation index (F*_ST_*) and the number of migrants per generation (Nm) (Table 2). The AMOVA indicated that 30.8% of the variance was due to the differences between structure groups, while 27.7% of the variance was between samples within structure groups. The majority of the variation was found within individuals (41.5%).

Pairwise population F*_ST_* values between different groups of accessions, sources of collection and biological races are presented in Table 3. From the results in Table 3, the structure groups had the highest F*_ST_* value G3 and G4 (0.23), and G1 and G4 had the lowest (0.13). For the sources of collection F*_ST_* values, the highest was recorded between IAR and ACCI (0.11) and the lowest between ACCI and ICRISAT (0.07). The highest F*_ST_* values ranged from 0.04 for Caudatum and Durra and 0.01 for Guinea and bicolor. Overall, the F*_ST_* estimates, averaged 0.18, 0.09 and 0.02 for structure groups, sources of collection and races respectively indicated that there is moderate genetic differentiation. The average F*_ST_* value (0.18) among the structure group was less than 0.25 which revealed the possibility of migration among the accessions. The result was confirmed by the Nm average value of 1.14 which revealed that there was enough gene flow and no clear partitioning of levels of genetic exchange according to structure group. Similar results were obtained for groupings among the sources of collections (F*_ST_* = 0.09 and Nm = 2.60) and the biological race (F*_ST_* = 0.02 and Nm = 16.58).

## 4. Discussion

Crop improvement depends on access to new sources of genetic variation. The most notable sources of genetic variation are landraces, wild or semi-wild relatives of cultivated crop species. In Africa, smallholder farmers produce the bulk of sorghum crops, mostly using unimproved landrace varieties and this enhance their resilience to climate variability. Landraces are valued for their beneficial genetic traits as local farmers have favored them for their ability to adapt to various environmental challenges [13]. To effectively harness these genetic resources, it is essential to understand and characterize the local germplasms. Assessing the genetic diversity and population structure of sorghum landraces is important for heterotic grouping, breeding population development, cultivar development and release [39].

The current study examined the genetic diversity present among 200 sorghum accessions, including landraces obtained from Nigeria using DArT markers. The polymorphic information content (PIC) provides an estimate of the information content of a marker. In this study, the highest PIC value observed was 0.38, indicating the presence of alleles in approximately 14.1% of the population. The average PIC value of 0.26 is similar to the findings of Afolayan et al. [13] and Enyew et al. [40] who used SNP markers to analyze sorghum germplasm collections. The findings of this study suggest that the SNP markers employed were sufficient in providing valuable information for assessing the extent of genetic diversity within the 200 examined sorghum accessions. The average observed heterozygosity (Ho) value of 0.15 obtained in this study aligns with the findings of Afolayan et al. [13], who used SNP markers for sorghum analysis (Ho = 0.22). However, it significantly surpasses the results from previous studies conducted by Enyew et al. [33] using SNP markers. The Ho value recorded was expected since sorghum is predominantly a self-pollinating crop, as noted by Sleper and Poehlman [41].

Heterozygosity is a fundamental measure of genetic variation in a population. The gene diversity (GD) of a locus, also known as its expected heterozygosity (He) describes the expected proportion of heterozygous genotypes under Hardy-Weinberg equilibrium [42]. In this study, the gene diversity of the SNP markers exhibited a range of 0.1 to 0.50 across all accessions, with an average of 0.32, indicating a high level of diversity. These informative markers can be effectively utilized for genotyping populations in genetic diversity studies, as suggested by Salem and Sallam [43]. In addition, Furthermore, the noticeable disparity between the observed heterozygosity (0.15) and the expected heterozygosity (0.32) values, and the relatively higher number of pairwise individuals with low genetic distance observed in this study indicates a limited genetic variation among the sorghum accessions. Another possible explanation may be that small-scale farmers frequently rotate the relevant landrace each year, employing rigorous selection criteria such as rainfall duration, panicle size, and plant aspects [44]. However, similar findings indicating a deficiency in heterozygosity have been reported in previous studies conducted by Motlhaodi et al. [45] and Enyew et al. [40].

The population structure analysis provides insights into the genetic diversity among sorghum genotypes and is useful in controlling false-positive associations between marker loci and traits of interest [46]. The results of Structure and PCoA analyses indicated a genetic structure comprised of four sub-populations of the sorghum accessions under study. The structure analysis did not display any pattern reflecting geographic adaptation. Clusters I and III were dominated by landraces grown by farmers in West Africa that were mostly tall, late maturing, adaptable and relatively high yielding. All the accessions from ACCI and some improved cultivars from ICRISAT and IAR were distinctly placed in Clusters II based on their relatedness in terms of early maturity, dwarf height and tolerance or susceptibility to drought. The grouping in cluster II suggested that they shared a common ancestry. The collections from ACCI were obtained from ICRISAT, while most of the breeding lines from IAR are a mixture of indigenous landraces and elite breeding lines obtained from ICRISAT. An intrinsic genetic subpopulation was visible for cluster 4, which included accessions from Nigeria obtained from IAR. The accessions are generally landraces grown by local farmers, and the cluster had the highest F_ST_ value (0.68). The SNP data showed that the test accessions had high ancestry membership coefficients of more than 0.60. The grouping of accessions from various collection areas together, despite their diverse origins, indicates a strong genetic association. These findings suggest that sorghum landrace genotypes are likely exchanged among regions by farmers, possibly through multiple routes. This aligns with the hypothesis of seed mixing, exchanging, and trade among small-scale farmers, highlighting the dynamic nature of genetic interactions and seed movement in agricultural communities.

Sorghum breeding efforts in West Africa were initiated in 1966 through the introduction of exotic lines. Subsequent pedigree breeding programs utilized local and exotic crosses, resulting in the release of improved pure line varieties and hybrids in the region [39]. Analysis using heatmap/dendrogram revealed four clusters that were consistent with the population structure analysis, indicating broad genetic variation among the 200 sorghum accessions studied. In contrast to the population structure analysis, the clusters identified through the dendrogram analysis exhibited partial alignment with geographical localization such as those from ACCI (AS 152, AS 1, and AS 66) and recent improved cultivars obtained from IAR (SAMSORG 44, 45, 46, and 49). The ACCI and IAR collections originated from introduced landraces from Sudan and shared early maturing and relatively drought-tolerant characteristics. Furthermore, nine elite breeding lines from ICRISAT formed a distinct cluster, indicating their high differentiation from other accessions, which is important for crop improvement purposes. In contrast, previous studies successfully clustered accessions based on geographic origins and racial groups [12,47]. Geographically isolated locations with limited interaction may become genetically distinct over time due to inbreeding. However, tracing such distinctiveness becomes challenging as germplasm movement frequently occurs across regions, facilitated by organizations like ICRISAT and ACCI.

Analysis of molecular variance in this study indicated that the genetic variation within populations (41.5%) was higher than that among populations (30.8%) and between samples within the structure (27.7%). In self-pollinating species like sorghum, the usual pattern is to maintain genetic variation within populations, while genetic variation tends to be lower among populations. This observation is consistent with previous studies that investigated genetic diversity using SNP markers [13] and SSR markers [48]. These studies also found higher genetic variation within sorghum accessions compared to the variation observed among the accessions, indicating that the accessions are not experiencing significant selection pressures. Nevertheless, a recent genetic diversity study utilizing SNP markers on sorghum accessions from Ethiopia revealed that 64.5% of the total variation was attributed to the variation among accessions, while 35.5% was attributed to the variation within accessions [40]. Similarly, in a study by Motlhaodi et al. [45] involving 22 sorghum accessions, a substantial genetic variation of 66.9% was observed among the accessions, with within-accession variation accounting for 23.6% of the total variation. The low genetic variation within the accessions is anticipated in self-pollinating crops, such as sorghum, as noted by Hamrick [49]. Furthermore, the high genetic variation within the population could be attributed to the preservation of sorghum landraces by farmers in Africa and suggested differences in adaptation and parentage. Genetic differentiation (F_ST_) quantifies the extent of genetic diversity resulting from allele frequency variations among populations, thereby reflecting population structure [50]. Values above 0.15 are considered significant in distinguishing populations, while values below 0.05 indicate a lack of substantial genetic structuring [51]. In our study, the highest F_ST_ value (0.23) was observed between subpopulations 3 and 4 (Table 3), indicating a notable genetic differentiation between these two subpopulations. Additionally, a substantial F index was identified within the groups derived from the three collection sources (ACCI, IAR, and ICRISAT), particularly between IAR and ACCI (0.11), indicating a pronounced genetic differentiation between these two groups. Furthermore, the F value for ACCI and ICRISAT (0.07) suggests a lower genetic variability. Although, overall, the F values among the three regions exhibit a continuous pattern, the observed degree of differentiation suggests a moderate gene flow between populations. This could potentially be attributed to seed exchange practices among neighboring farmers, as it is well-known that local farmers exchange seeds to enhance crop productivity. The Gene flow (Nm) value between subpopulation 2 and 4 (N_m_ = 0.994) was low suggesting that a low genetic exchange might occur which led to a high genetic differentiation (0.201) between the subpopulations (Table 3). Accessions in subpopulation 2 were mostly comprised of materials collected from ACCI and ICRISAT while subpopulation 4 comprised of local landraces adapted to Nigeria and obtained from IAR. There is a possibility that low gene flow between the subpopulations can be because of isolation from gene exchange by distance or due to small population size in the study. High Nm value of 1.616 was shown by subpopulation 1 and 4, revealing a possibility of high genetic exchange. This suggests subpopulation 1 and 4, which are landraces and elite breeding lines from Nigeria collected from IAR and ICRISAT may have had common ancestry. According to Wright [37], an Nm value less than one indicate limited gene exchange among subpopulations. The result of the study reveals sufficient genetic variability in the sorghum accessions, which could be useful in sorghum breeding programs. The germplasm collections were composed mainly of landraces, which are known to be highly heterogeneous. Previous studies from Nigeria [13], Ethiopia [48], South Africa [52], Burkina Faso [53] and Cameroon [54] have documented the existence of large genetic variation within landrace collections of sorghum in Africa.

## 5. Conclusions

The present study reports genetic diversity studies and population structure analysis on a panel of 200 sorghum germplasm collections of West African origin, using DArT-Seq derived SNP markers, as a basis for future breeding. The SNP markers employed in this study exhibited a considerable degree of polymorphism, effectively revealing the genetic differences between and within the sorghum populations. Approximately half of the SNP markers were highly informative, making them valuable candidates for future genetics studies. Interestingly, a notable proportion of loci displayed an excess of heterozygosity. Exploring these loci further by studying genotypes with different alleles could provide insights into their relevance and importance in terms of desirable traits. The significant genetic differentiation observed among the sorghum accessions stemming from diverse germplasm collections across Africa will be valuable for sorghum breeders in identifying and selecting desirable parent plants for effective hybrid breeding strategies. Therefore, future sorghum improvement should focus on genetic improvement using the landraces. The formation of four distinct clusters among the sorghum accessions highlights the potential for crossbreeding genotypes from different clusters to assess their progenies for desirable traits. The study identified distantly related sorghum accessions such as SAMSORG 48, ICNSL2014-024-2, KAURA RED GLUME (Cluster I); Gadam, AS 152, AS 1, MACIA (Cluster 2); CSRO1, CAPARLASG2015002, ICNSL2014-062 (Cluster 3); and YALAI, KAFI MORI, FARA DOGON DAWA (Cluster 4). The distantly related sorghum accessions will be used in creating new gene pools and novel genotypes for sorghum breeding programs in Nigeria and similar agro-ecologies in Africa. Although the SNP markers used in this study adequately discriminated between the accessions, it is important to conduct phenotypic evaluations to fully elucidate the genetic basis of phenotypic variation for crop improvement.

## Figures and Tables

**Figure 1 genes-14-01480-f001:**
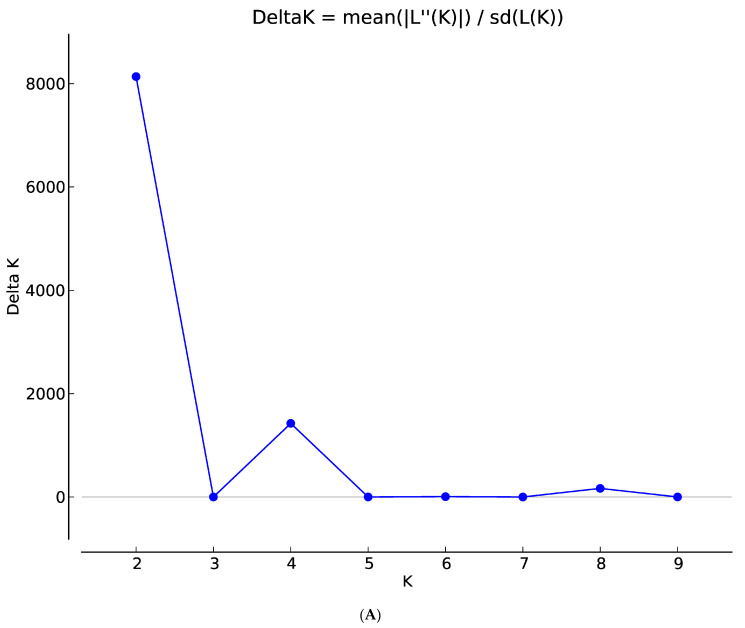
(**A**) Graph of estimated membership fraction based on Structure Analysis. The maximum of adhoc measure ΔK determined by structure harvester was found to be K = 4, which indicated that the entire population could be grouped into four clusters. (**B**) The Structure Plot for K = 4 at individual and across iteration of 200 sorghum accessions based on 7516 DArT SNP markers. Values in the *y*-axis show coefficient of membership/assignment. Each coloured segment per genotype estimates the membership fraction to each of the four sub-populations (G1 = pink, G2 = turquoise blue, G3 = green, G4 = purple).

**Figure 2 genes-14-01480-f002:**
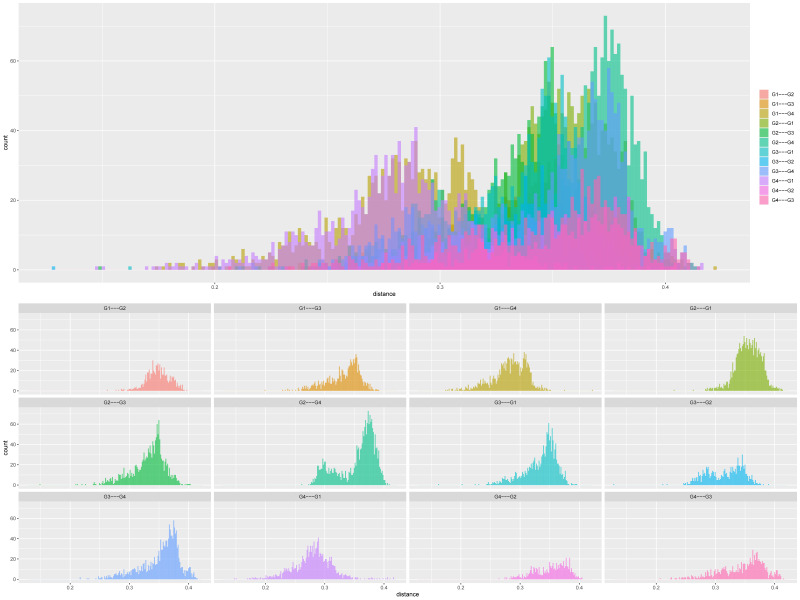
Distribution of genetic distances (GD) obtained using DArT SNP markers based on structure group for 200 sorghum accessions (G1 = pink, G2 = turquoise blue, G3 = green, G4 = purple). In all panes, the *x*-axis displays the distance, and the *y*-axis displays the count/percentage of the pairs found at that distance.

**Figure 3 genes-14-01480-f003:**
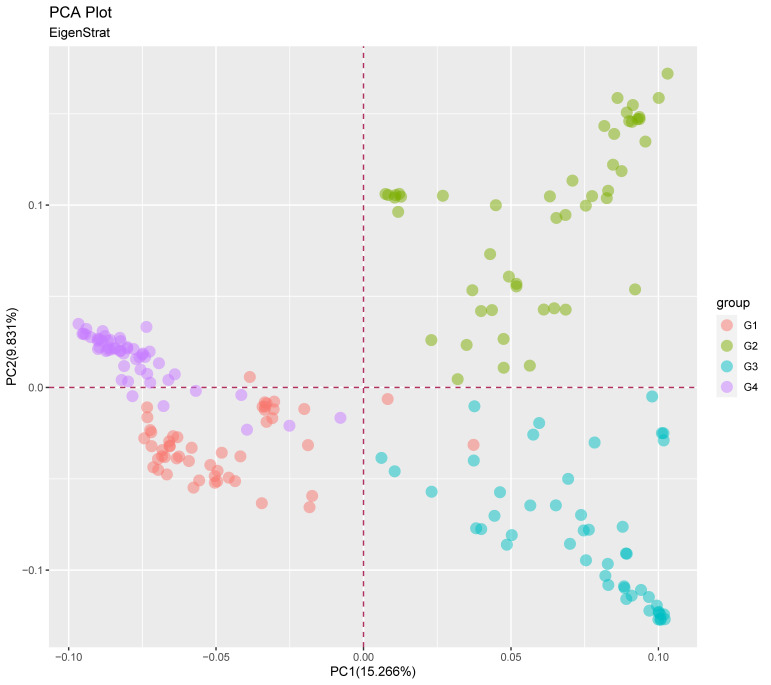
Principal coordinate analysis (PCoA) plot of the 200 sorghum accessions based on the 7516 DArT SNP markers. PC1 and PC2 are the first and second principal coordinate, respectively, and number in parentheses refer to the proportion of variance explained by the principal coordinates. (G1 = pink, G2 = turquoise blue, G3 = green, G4 = purple).

**Figure 4 genes-14-01480-f004:**
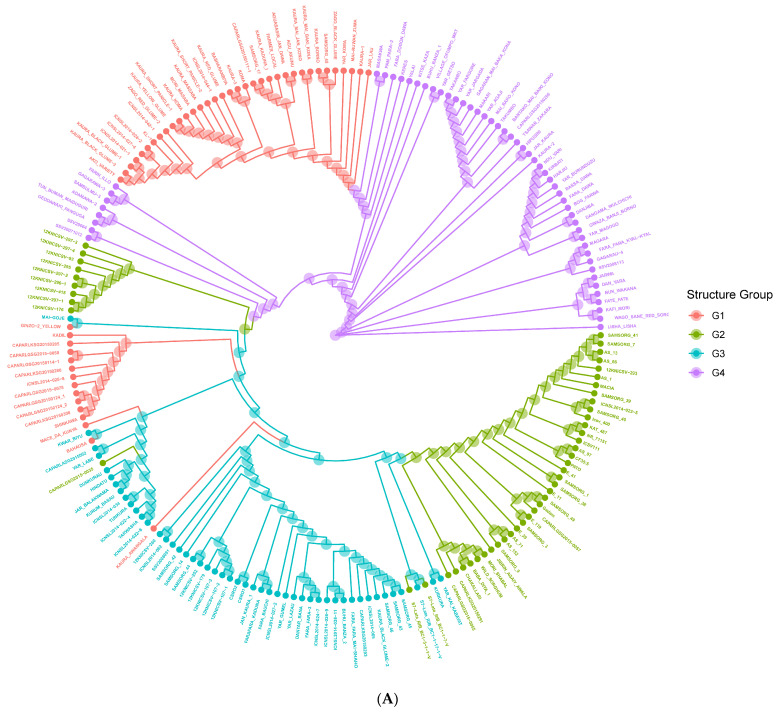
Neighbor Joining Clustering based on Roger’s Distance (Cladogram View) for 200 sorghum accessions. Colors are assigned based on (**A**) population structure (G1 = red, G2 = green, G3 = blue, G4 = purple) or (**B**) Race (Bicolor = red, Caudatum = yellow, Durra = green, Guinea = blue, Kafir = purple).

**Table 1 genes-14-01480-t001:** Summary statistics of diversity indices for 200 sorghum accessions based on 7516 SNP markers.

	Genetic Parameters				
Statistics	GD	Ho	PIC	MAF	MaF
mean	0.32	0.15	0.26	0.23	0.77
lower	0.10	0.01	0.09	0.05	0.50
upper	0.50	0.79	0.38	0.50	0.95

GD = Gene diversity, Ho = Observed heterozygosity, PIC = Polymorphic information content, MAF = Minor allele frequency, MaF = Major allele frequency.

**Table 2 genes-14-01480-t002:** Analysis of molecular variance among and within four subpopulations of 200 sorghum accessions evaluated based on 7516 SNP markers.

Source of Variation	Degree of Freedom	Sum Square	Mean Square	Components of Covariance	Proportion of Variance (%)	*p* Value
Between Structure Groups	3	247,342.6	82,447.5	799.7	30.8	0.0010
Between samples Within Structure Groups	196	493,070.7	2515.7	719.9	27.7	0.0010
Within samples	200	215,176.0	1075.9	1075.9	41.5	0.0010
Total	399	955,589.3	2395.0	2595.5	100.0	

**Table 3 genes-14-01480-t003:** Pairwise F_ST_ matrix among four subpopulations of 200 sorghum accessions evaluated based on 7516 SNP markers [Note: values in top diagonal shows gene flow (Nm), while bottom diagonal values are genetic differentiation (F*_ST_*)].

Gene Flow (Nm)														
	G1	G2	G3	G4		ACCI	IAR	ICRISAT		Kafir	Bicolor	Caudatum	Durra	Guinea
G1	0.000	1.026	1.026	1.616	ACCI	0.000	2.044	3.321	Kafir	0.000	14.036	11.166	5.024	26.066
G2	0.196	0.000	1.342	0.994	IAR	0.109	0.000	2.438	Bicolor	0.018	0.000	11.542	9.79	40.734
G3	0.196	0.157	0.000	0.861	ICRISAT	0.070	0.093	0.000	Caudatum	0.022	0.021	0.000	5.619	22.686
G4	0.134	0.201	0.225	0.000					Durra	0.047	0.025	0.043	0.000	19.13
									Guinea	0.01	0.006	0.011	0.013	0
Genetic differentiation (F*_ST_*)														

## Data Availability

All of the material is owned by the authors and/or no permissions are required. The data that support the findings of this study are available from the corresponding author upon reasonable request.

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
