# Peer review of "Genetic Diversity and Population Structure of African Sorghum (Sorghum bicolor L. Moench) Accessions Assessed through Single Nucleotide Polymorphisms Markers"

_genes, 2023, doi:10.3390/genes14071480_

Round 1

Reviewer 1 Report

The study is presented well and is of sound importance. I find it interesting and there are only few minor points as below that can be improved:

* Provide details of the DNA extraction protocol instead of just citing.

* Please write all the R packages used in the study, authors have reported only few, all packages should be reported along with the analysis conducted through those packages. So that readers will be able to reproduce the results.

* Figures quality should be improved. Especially Figure 4

* please mention what is on the x and y axis in the figure 2. 

Author Response

14 July 2023

Dr. Zona Wang
Section Managing Editor, MDPI Genes,

Submission of a revised manuscript ID genes-2474374.R1

Thank you for the feedback on our manuscript with ID genes-2474374.R1 Entitled: “Genetic diversity and population structure of African sorghum (Sorghum bicolor L. Moench) accessions assessed through single nucleotide polymorphisms markers”. We are grateful for the valuable contributions of the reviewers which have further enriched the manuscript.

We have carefully made the required revisions which are indicated using track changes and highlighted text in the present submission as shown in blue font text. Please find Below point-by-point response and amendments carried out following the reviewers’ and editor’s comments:

RESPONSE TO REVIEWER 1

Comment 1:  Provide details of the DNA extraction protocol instead of just citing.

Response 1: Thank you very much for taking the time to review our manuscript. We have now carefully paid attention to all the queries and comments raised to improve the manuscript.  The DNA extraction protocol has been provided in detail.

The paragraph now reads as follows -

DNA extraction and genotyping-by-sequencing (GBS)

The genotypes were grown in a plant growth chamber (Conviron, Canada) at the Biosciences eastern and central Africa-International Livestock Research Institute (BecA-ILRI) hub using cell trays. Three seeds of each genotype were sown per tray. Three-week-old leaf sample was collected from the three seedlings and the pooled leaf samples were frozen in liquid nitrogen and stored at −80°C for later use. Genomic DNA (gDNA) was extracted from the frozen tissue according to the CTAB protocol, with some modifications (21). The quantity of extracted DNA in each sample was determined using a Thermo Scientific NanoDrop Spectrophotometer 2000c. The quality of the extracted DNA was checked on 0.8% agarose gel run in 1% TAE buffer at 70 V for 45 minutes. After the quality had been checked, 40 μl of a 50 ng/μl gDNA of each sample of the 200 sorghum lines was sent for whole genome scanning using Genotyping by sequencing (GBS) technology as described by Elshire et al. (22), using DArTseqTM technology (https://www.diversityarrays.com/) of the Integrated Genotype Service and Support (IGSS) platform in Nairobi, Kenya. The GBS was performed by using a combination of DArT complexity reduction methods and next generation sequencing following protocols described in (22 - 24).  Marker development was based on the protocol of Elshire et al. [22], using the ApeKI restriction enzyme (recognition site, G| CWCG). Reads and tags found in each sequencing result were aligned to the sorghum reference genome v2.1 (available via https://phytozome.jgi.doe.gov/pz/portal.html#!info?alias=Org_SbicolorRio_er). Each allele was scored in a binary fashion ("1" = Presence, "0" = Absence and '-'for failure to score) while heterozygotes were scored as 1/1 (presence for both alleles/both rows).

Comment 2: Please write all the R packages used in the study, authors have reported only few, all packages should be reported along with the analysis conducted through those packages. So that readers will be able to reproduce the results.

Response 2: The R packages used in the study have been revised and explained in detail.

The section now reads as follows:

Genetic diversity parameters: polymorphic information content (PIC), minor allele frequency (MAF), major allele frequency (MaF), observed heterozygosity (Ho), and expected heterozygosity or gene diversity (GD) were estimated for each defined group with R software [23].

Population structure analysis

Two complementary methods were employed to deduce the population structure of the 200 African sorghum accessions. The first method involved using a Bayesian model-based clustering algorithm called STRUCTURE software (59). The second method employed principal coordinate analysis (PCoA). Bayesian-based clustering was performed using STRUCTURE software v.2.3.4 [24] with four independent runs with K from 2 to 10, each run with a burn-in period of 10,000 iterations and 50,000 Monte Carlo Markov iterations, assuming the admixture model. The output was subsequently visualized using the web-based program Structure Harvester v.06.94 [25] and the number of clusters was inferred according to the Evanno method [26]. The software CLUMPP version 1.1.2 (62) was utilized to align the cluster assignments obtained from separate runs. The input files generated by structure Harvest were employed for this purpose. Bar plots were created using DISTRUCT version 1.1 (63) to visualize the average outcomes of the runs, focusing on the most likely value of K. A genotype was considered part of a particular group if its membership coefficient was ≥ 0.70. Genotypes with membership coefficients below 0.70 for each assigned K were considered to be admixed. Phylogenetic relationships between the lines were inferred using the unweighted neighbour-joining method [27] and plotted using R software [23] based on Rogers' dissimilarity [28]. Heatmap of the genetic distance value among lines was generated using gplots R package. The adegenet R package [29] was used to extract and plot pairwise distances between different groups identified from structure analysis. PCoA (Principal Coordinates Analysis) is a method that uses distances to analyze and visualize differences between individuals. The Eigenstrat method [30] based on principal components analysis was used to study population relationships further, and two-dimension PCA plot was generated using ggplot2. In order to compare the number of clusters obtained from STRUCTURE, a population genetic model, with the clusters obtained from PCoA, the dartR-R package was employed. This comparison was made without making any assumptions about the underlying genetic model of the population (65).

The number of subpopulations determined with STRUCTURE was used for analysis of molecular variance (AMOVA) and the calculation of Nei's genetic distance (Excoffier et al., 1992) using poppr package in the R version 2.8.3 (Kamvar  et a., 2015; Pembleton et al., 2013; 23]. From AMOVA, the fixation index (FST) and Nm (haploid number of migrants) within the population were obtained. FST measures the amount of genetic variance that can be explained by population structure based on Wright's F-statistics [31], while gene flow was estimated using an indirect method based on the number of migrants per generation (Nm) as (1- FST)/4 FST.

The phylogenetic relationships of the subpopulations were generated based on pair-wise fixation indexes using the StAMPP package (Pembleton et al., 2013) and neighbor joining trees were constructed using the dartR package in R. The dissimilarity or distance matrix, based on pair-wise genetic frequencies, was computed using the Euclidean distance formula in the R environment. The resulting dissimilarity matrix was used to create a tree structure using the unweighted pair group method analysis (UPGMA) algorithm, which was performed using the ggdendro and ggplot2 packages within the same software. Phylogenetic trees were constructed in R using the hclust algorithm, and the UPGMA agglomeration method was employed to determine the relevant clustering of data points.

Comment 3:  Figures quality should be improved. Especially Figure 4

Response 3:  The quality of the figures has been improved and included in the manuscript.

Comment 4: please mention what is on the x and y axis in the figure 2.

Response 4 : The caption for Figure has been revised and it reads as follows:

Figure 2: Distribution of genetic distances (GD) obtained using DArT SNP markers based on structure group for 200 sorghum accessions (G1 = pink, G2 = turquoise blue, G3 = green, G4 = purple). In all panes, the x-axis displays the distance, and the y-axis displays the count/percentage of the pairs found at that distance.

With best regards.

Muhammad Ahmad Yahaya

Reviewer 2 Report

Dear editor;

The study evaluates the genetic diversity and population structure of a collection of 200 African sorghum accessions using SNP markers. The methodology is appropriate to meet the objectives. The results are novel and in accordance with the proposed objectives. The discussion is coherent. The paper should take into account some minor comments.

Line 3: species name in italics

Line 130:

Line 169: Analyze genetic diversity by genetic group according to structure. Include nucleotide diversity parameter (pi).

Line 177- 178: indicate the membership threshold to assign an individual to a genetic group, when they were considered admixtures?

Line 244: Improve the resolution of Figure 4.

Line 272: Improve the table. It would be better to show the results in separate tables or with a figure.

Author Response

14 July 2023

Dr. Zona Wang
Section Managing Editor, MDPI Genes,

Submission of a revised manuscript ID genes-2474374.R1

Thank you for the feedback on our manuscript with ID genes-2474374.R1 Entitled: “Genetic diversity and population structure of African sorghum (Sorghum bicolor L. Moench) accessions assessed through single nucleotide polymorphisms markers.” We are grateful for the valuable contributions of the reviewers, which have further enriched the manuscript.

We have carefully made the required revisions which are indicated using track changes and highlighted text in the present submission as shown in blue font text. Please find Below point-by-point response and amendments carried out following the reviewers’ and editor’s comments:

RESPONSE TO REVIEWER 2

Comment 1:  Line 3: species name in italics

Response 1: Thank you very much for taking the time to review our manuscript. We have now carefully paid attention to all the queries and comments raised to improve the manuscript.  The MDPI Genes template was used to draft the manuscript. The article name was italicized by the template. The academic editor will ensure the article title is in the right format before publication.

Comment 2:  Line 169: Analyze genetic diversity by genetic group according to structure. Include nucleotide diversity parameter (pi).

Response 2:  The genetic diversity by genetic group according to structure was not analysed and added to the submitted manuscript due to time factor. The co-authors responsible for the analysis are currently relocating from ICRISAT India to CIMMYT Kenya.

Comment 3: Line 177- 178: indicate the membership threshold to assign an individual to a genetic group, when they were considered admixtures?

Response 3: The membership threshold statement has been revised and a figure assigned in the materials and method section.

Line 162 and 163 reads as follows:

A genotype was considered part of a particular group if its membership coefficient was ≥ 0.70. Genotypes with membership coefficients below 0.70 for each assigned K were considered to be admixed.

Comment 4: Line 244: Improve the resolution of Figure 4

Response 4: The quality of the figures has been improved and included in the manuscript.

Comment 5: Line 272: Improve the table. It would be better to show the results in separate tables or with a figure.

Response 5: The results in Table 3 has been improved.

With best regards.

Muhammad Ahmad Yahaya
